# Second-line systemic treatment for metastatic colorectal cancer: A systematic review and Bayesian network meta-analysis based on RCT

**Chengyu Sun[1]\*, Enguo Fan[2], Luqiao Huang[1], Zhengguo Zhang[1]\***

**1** Department of Colorectal Surgery, The Affiliated Xuzhou Clinical College of Xuzhou Medical University, Xuzhou Central Hospital, Xuzhou, Jiangsu, China, **2** State Key Laboratory of Medical Molecular Biology, Department of Microbiology and Parasitology, Institute of Basic Medical Sciences Chinese Academy of Medical Sciences, School of Basic Medicine Peking Union Medical College, Beijing, China

\* Scy17852072130@163.com (CS); ZGZhangXZ@163.com (ZZ)

## Abstract

### Background

The optimal second-line systemic treatment for metastatic colorectal cancer (mCRC) is inconclusive.

### Methods

We searched PubMed, Web of Science, EMBASE, and Cochrane Library for RCTs comparing second-line systemic treatments for mCRC from the inception of each database up to February 3, 2024. Markov Chain Monte Carlo (MCMC) technique was used in this network meta-analysis (NMA) to generate the direct and indirect comparison results among multiple treatments in progression-free survival (PFS), overall response rate (ORR), overall survival (OS), complete response (CR), partial response (PR), grade 3 and above adverse events (Grade $\geq$ 3AE), and any adverse events (Any AE). The surface under the cumulative ranking curve (SUCRA) was adopted to evaluate the probability of each treatment being the optimum intervention. Subgroup analyses were performed based on the RAS gene status.

### Results

A total of 47 randomized controlled trials were included, involving 16,925 patients and 44 second-line systemic treatments. In improving OS, FOLFOX + Bevacizumab + Erlotinib exhibited significant superiority (SUCRA:92.7%). In improving PFS, Irinotecan + CMAB009 (SUCRA:86.4%) had advantages over other treatments. FOLFIRI + Trebananib (SUCRA:88.1%) had a significant advantage in improving ORR. Among multiple second-line treatments, the SUCRA values of FOLFOX + Bevacizumab in PFS, OS, ORR, and PR were 83.4%, 74.0%, 81.1%, and 86.1%, respectively, and the safety was not significantly different from other interventions. Subgroup analyses showed that FOLFIRI + Bevacizumab + panitumumab ranked among the top in survival outcomes in the RAS-mutant population

**Data Availability Statement:** All data are in the manuscript and supporting information files.

**Funding:** This study has received funding by Technology Program of Xuzhou Municipal Health

and Wellness Commission (KC21245). The funders had no role in study design, data collection and analysis, decision to publish, or preparation of the manuscript.

**Competing interests:** The authors have declared that no competing interests exist.

(OS SUCRA: 87.9%; PFS SUCRA: 70.2%); whereas in the RAS-wild-type population, FOL-FIRI + Bevacizumab significantly improved survival outcomes (OS SUCRA: 73.2%; PFS SUCRA: 65.1%).

## Conclusion

For most people, FOLFOX + Bevacizumab may be the best second-line systemic treatment regimen for mCRC. For RAS-mutant populations, FOLFIRI + Bevacizumab + Panitumumab is recommended. However, the therapeutic effect may be affected by the patient's physiological state, and clinicians should apply it based on actual conditions.

## 1. Introduction

Colorectal cancer (CRC) has become the third most prevalent cancer and the second leading cause of death worldwide. More than 1.9 million new cases of CRC and 935 thousand deaths were reported in 2020 [1]. It is estimated that by 2030, the global burden of the disease will increase by 60%, with more than 2.2 million new cases and over 1.1 million deaths [2]. About 20%-25% of patients with CRC have distant metastasis at the time of initial diagnosis, and 30% of the patients develop in situ recurrence or distant metastasis during the course of the disease [3, 4]. Although surgery can improve the 5-year and 10-year survival rates of these patients by 38% and 26%, respectively, only a small number of advanced CRC patients are suitable for surgical treatment [5–7], and for unresectable, metastatic colorectal cancer (mCRC), systemic therapy is an important and feasible approach [8].

First-line treatments for mCRC are the key determinants of treatment outcomes with the longest duration of treatment [9]. The classic first-line treatments include the combination of fluoropyrimidine-based cytotoxic drugs, for example, the dual regimen like FOLFIRI: folinic acid, 5-FU, and Irinotecan, FOLFOX: folinic acid, 5-FU, and Oxaliplatin, as well as the triple regimen like FOLFOXIRI: folinic acid, 5-FU, Oxaliplatin, Irinotecan and cytotoxic drugs combined with epidermal growth factor receptor inhibitor (anti-EGFR: Cetuximab and Panitumumab) or vascular endothelial growth factor inhibitor (anti-VEGF: Bevacizumab, Axitinib and Aflibercept). The tumor location and biomarkers (RAS, BRAF, etc.) should be considered before selecting a first-line treatment [9, 10]. If a patient experiences disease progression or relapse during the treatment cycle but has good clinical performance and adequate organ function, he or she will require second-line therapy, which is used after first-line therapy due to progression or unacceptable toxicity, with the goal of prolonging overall survival (OS) or progression-free survival (PFS). Therefore, the efficacy and toxicity of drugs are key factors in determining treatments [10]. Commonly used second-line treatments are cytotoxic drug therapy alone or monoclonal antibody therapy in addition to the first-line treatment, and studies have shown that the median response rate and PFS of FOLFOX are significantly higher than those of FOLFIRI [11] in second-line settings. The clinical value of Bevacizumab, Cetuximab, Panitumumab, Trebananib, and other monoclonal antibodies have also been validated in research [12–15]. However, no up-to-date guidelines compare all available second-line treatments, and the optimal clinical second-line regimen is unclear. Similar studies exploring the advantages and disadvantages of second-line treatment options for mCRC have been published [16–19]. However, a comprehensive and intuitive comparison is lacking, which poses certain challenges to the development of this field. Hence, we conducted this systematic review and meta-analysis to comprehensively and systematically explore the effectiveness and safety

of all second-line treatment options for mCRC currently used in clinical practice, and try to identify the best options.

## 2. Materials and methods

This systematic review with NMA was reported in accordance with the Preferred Reporting Items for Systematic Review incorporating Network Meta-Analysis (PRISMA NMA) (S5 File) [20, 21] and the Cochrane Collaboration Handbook. Moreover, the study protocol was previously registered in PROSPERO (No. CRD42023433935).

### 2.1 Database and search strategy

Two investigators independently searched for all articles on second-line treatments for mCRC published up to February 3, 2024. This review followed the PRISMA-NMA guideline [20]. The selected databases included PubMed, Web of Science, EMBASE, and Cochrane Library. The specific search strategies are shown in Table 1 in S1 File.

### 2.2 Selection of studies and eligibility criteria

Studies that met the following criteria were included:

(1) Population: the subjects of the study were mCRC patients; (2) Intervention and control: a second-line treatment was used in both the study and control groups, and the drug types, dosages and order of medication must be specified for the second-line and first-line treatment regimens; (3) Outcome: the reported results included progression-free survival (PFS), overall survival (OS), overall response rate (ORR), complete response (CR), partial response (PR), grade 3 and above adverse events (Grade $\geq$ 3AE), and some or all of any adverse events (Any AE); (4) Study design: the study type was a randomized controlled trial (RCT). If more than one article reported the same population, the one with more complete data was included.

The following studies were excluded: (1) animal experiments; (2) duplicated publications; (3) meta-analysis, reviews, conference abstracts, case reports, and guidelines; (4) the full text of a study was unavailable even after multiple searching attempts.

### 2.3 Outcomes

The primary outcome measures were OS, PFS, ORR, and Grade $\geq$ 3AE. The secondary observation indicators were CR, PR, and Any AE. The OS was defined as the time period from the date of randomization grouping to death from any cause. The definition of PFS was the time from the date of randomization grouping to disease progression (PD). The ORR was defined as the number of patients with CR or PR confirmed by central radiological examination. Grade $\geq$ 3AE indicated the number of grade 3 to grade 5 adverse reactions during the process of medication in clinical trials; Any AE referred to the number of any adverse reactions during the process of medication in clinical trials and was usually graded according to the Common Terminology Criteria for Adverse Events (CTCAE).

### 2.4 Data extraction

A Microsoft Excel sheet was developed in advance to record data extracted from the included studies, and data review and extraction were conducted by two independent researchers, including first author, research country or region, year of publication, duration of study, sample sizes in the treatment and control groups, average age, tumor stage and type, interventions, previous first-line interventions, survival data by median PFS and OS, response data by ORR, CR, and PR, and safety data by Grade $\geq$ 3AE and Any AE. The hazard ratios (HRs) of PFS and

OS and corresponding 95% confidence interval (95%CI), ORR, CR, PR, Grade $\geq$ 3AE, Any AE were extracted, respectively. If the required data is incomplete, we tried to contact the lead author to obtain the data. Disagreements arising during this process were addressed through consultation with a third researcher.

### 2.5 Quality evaluation

The quality evaluation of the included studies was performed by two investigators using the risk of bias assessment tool RoB2.0 released by Cochrane [22], which was used to evaluate the possibility of bias in RCTs from five different aspects: (1) bias arising from the randomization process; (2) bias due to deviations from the intended interventions; (3) bias arising from missing outcome data; (4) bias arising from outcome measurement; (5) bias due to selective reporting of outcomes. Disagreements arising during this process were resolved through consultation with a third researcher.

### 2.6 Data analysis

StataSE 15.0 and R4.2.2 were used for plotting network diagrams and data analysis. The data of seven outcomes, like PFS, OS, ORR, CR, PR, Grade $\geq$ 3AE, and Any AE were analyzed using a hazard ratio (HR) and the 95% confidence interval (95%CI) for analysis of survival data (PFS, OS), relative risk (RR) for the statistical indicators of the effect of dichotomous data, and 95% CI for reporting the results of network comparison. No significant difference existed when the 95% CI contained 1, and significant differences were considered when the 95% CI did not contain 1. Given the heterogeneity between the included literature, a random effects model was used for analysis. Bayesian NMA was performed for each outcome, using Markov Chain Monte Carlo (MCMC) simulation technique. A network diagram was formed for each outcome, where different interventions are represented by different nodes, and the size of the nodes represents the number of subjects. Two nodes are connected by a weighted line segment, the thickness of which represents the amount of direct evidence available between them. Sensitivity analysis was carried out by comparing the deviation information criterion (DIC). DIC below five between the consistency test and the inconsistency test indicated that the model fitted well. Node-splitting method was used for plotting diagrams to compare the direct and indirect evidence. The surface under the cumulative ranking curve (SUCRA) was adopted to represent the probability of each intervention being the optimum intervention, and a larger SUCRA indicated that the corresponding intervention was more effective. Subgroup analyses of OS and PFS were performed in patients according to RAS types. Funnel plots were developed for detecting the publication bias of the included articles.

## 3. Results

### 3.1 Literature search and screening process

After a comprehensive and systematic search, a total of 32,909 studies were obtained, and 22,635 studies were obtained after duplicate studies were excluded. A total of 427 studies were obtained by reviewing the titles and abstracts (S2 File), and 47 studies were finally considered to meet the inclusion criteria after the full texts were searched and reviewed. The specific screening process of studies is shown in Fig 1.

### 3.2 Basic characteristics of included studies and risk of bias assessment

Basic characteristics of the 47 included RCTs [23–69] were summarized in S3 File and Table 2 in S1 File. One study compared four groups [25], each of five studies compared three groups

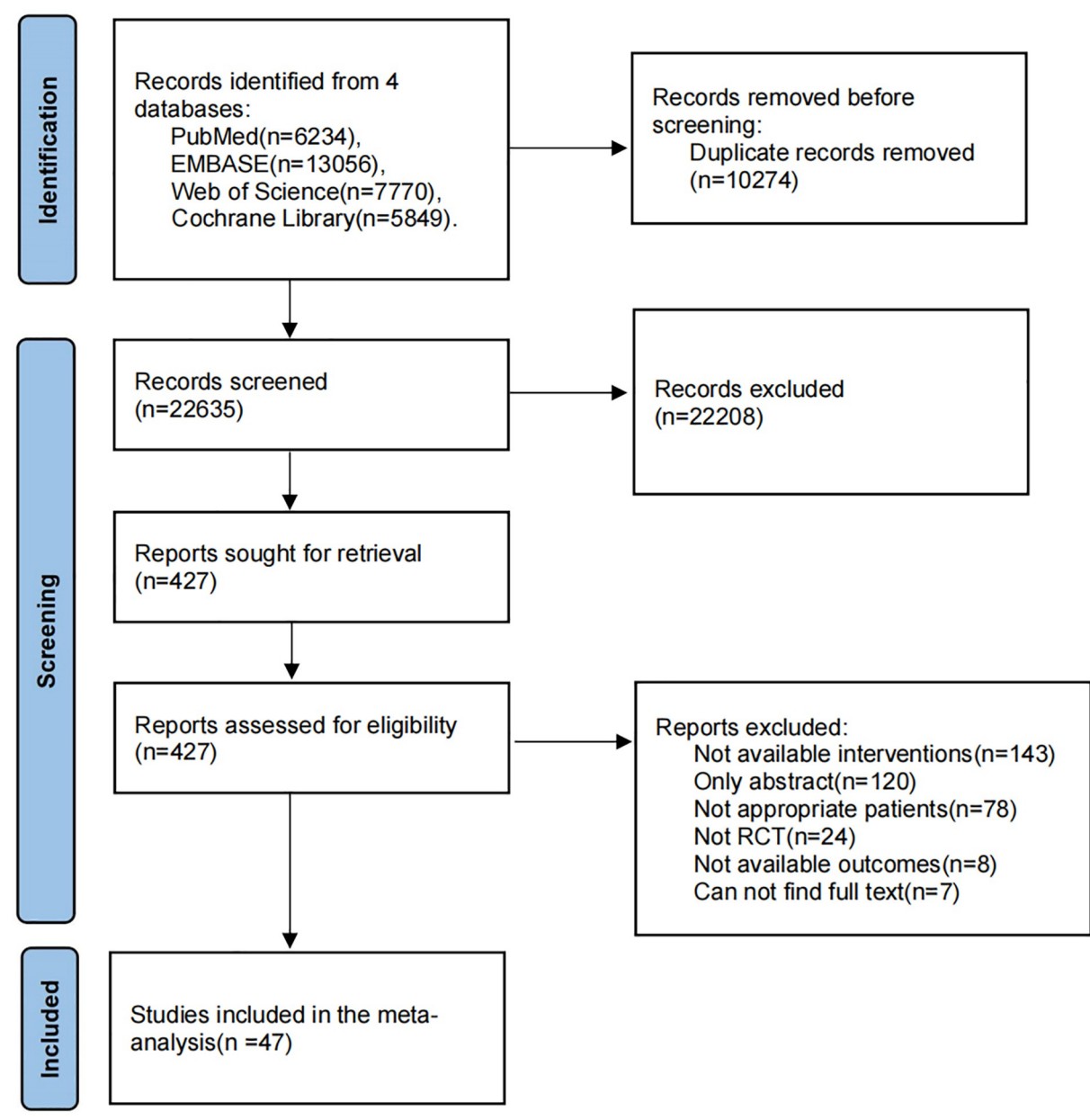

**Fig 1. PRISMA flow diagram for included studies.**

[29, 35, 45, 53, 54] and each of the remaining 41 studies compared two groups. The included studies covered 44 different interventions involving 16,925 patients, most of whom were men, and the median age was between 55 and 70 years old.

### 3.3 Risk of bias assessment

Most of the included studies in this review were large multi-center RCTs, and there were fewer biases in terms of outcome completeness, outcome measurement, and outcome reporting. Because only 12 of the included studies were double-blind experimental studies as clearly stated [29, 31, 33, 34, 39, 43, 48, 57, 62, 65–67], while others used open-label design or

unspecified types of blind design, most studies were at risk in random assignment and bias of interventions. Six of the included studies were rated as overall low risk [33, 34, 43, 48, 57, 64], three studies were rated as overall high risk [24, 27, 56], and other studies had certain risks. In general, the included studies were of average quality, and more rigorous RCTs were needed to supplement the results of this study (Figs 1 and 2 in S1 File; S4 File).

### 3.4 Network diagram for available evidence

Fig 2 shows the network diagrams of the seven outcomes like PFS, OS, ORR, CR, PR, Grade ≥ 3AE, and Any AE, and for each outcome, 41, 41, 40, 17, 36, 37, 16 second-line treatments were included in the final analysis, respectively. In the figure, the thickness of the lines is proportional to the number of literature comparisons, and the size of the diameter of the circles is proportional to the number of people treated with the corresponding intervention. The results showed that FOLFOX, FOLFIRI, and Irinotecan circles had larger diameters and more lines with the targeted drugs, suggesting a larger population of patients treated with classical chemotherapy regimens in combination with targeted drugs.

### 3.5 Inconsistency test

Fig 3 in S1 File shows the results of local inconsistency tests for all available outcomes. According to the analysis results using node-splitting method, the PFS, OS, ORR, PR, Grade ≥ 3AE, and Any AE all have P values > 0.05, indicating that there is no difference in direct or indirect comparisons between interventions, and no apparent local inconsistency among the available outcomes.

### 3.6 League table and SUCRA ranking for probability

**3.6 1 Overall survival.** According to the results of comprehensive comparison among interventions, FOLFOX + Bevacizumab + Erlotinib was better in improving OS compared with 5-FU (HR = 0.31[95%CI:0.13–0.75]), Bevacizumab (HR = 0.47[95%CI:0.23–0.96]), Cetuximab (HR = 0.29[95%CI:0.10–0.83]), FOLFIRI + Simtuzumab (HR = 0.36[95%CI:0.14–0.92]), FOLFOX (HR = 0.49[95%CI:0.25–1.00]), FOLFOX + Icrucumab (HR = 0.40[95%CI:0.17–0.99]), FOLFOX + Linifanib (HR = 0.44[95%CI:0.20–0.98]), Irinotecan (HR = 0.43[95%CI:0.19–0.97]), Irinotecan + Cetuximab (HR = 2.32[95%CI:1.04–5.35]), Panitumumab (HR = 0.29[95%CI:0.09–0.89]) and Supportive Care (HR = 0.26[95%CI:0.11–0.64]), with statistically significant difference as shown in Fig 4A in S1 File. According to SUCRA for probability, the top three were FOLFOX + Bevacizumab + Erlotinib (SUCRA: 92.7%), FOLFOX + Axitinib (SUCRA: 88.6%), and FOLFIRI + Panitumumab + Bevacizumab (SUCRA: 83.4%) (Fig 3A; Table 3 in S1 File).

**3.6.2 Progression-free survival.** According to the results of comprehensive comparison among the available interventions, Irinotecan + CMAB009 had an advantage in improving PFS over 5-FU (HR = 0.36[95%CI:0.20–0.63]), Bevacizumab (HR = 0.50[95%CI:0.25–0.97]), Capecitabine + Temozolomide (HR = 0.44[95%CI:0.22–0.92]), FOLFIRI (HR = 0.53[95%CI:0.32–0.90]), FOLFIRI + Napabucasin (HR = 0.51[95%CI:0.27–0.98]), FOLFIRI + Simtuzumab (HR = 0.38[95%CI:0.20–0.76]), FOLFIRI + Trebananib (HR = 0.43[95%CI:0.20–0.92]), FOLFOX + Icrucumab (HR = 0.42[95%CI:0.19–0.92]), Irinotecan (HR = 0.50[95%CI:0.33–0.76]), Irinotecan + S-1 (HR = 0.49[95%CI:0.26–0.97]) and Oxaliplatin (HR = 0.33[95%CI:0.18–0.61]), with statistically significant difference (Fig 4B in S1 File). SUCRA ranking for probability of all interventions indicated that the top three were Irinotecan + CMAB009 (SUCRA: 86.4%), FOLFOX + Bevacizumab (SUCRA: 83.4%) and FOLFOX + Nintedanib (SUCRA: 82.0%) (Fig 3B; Table 3 in S1 File).

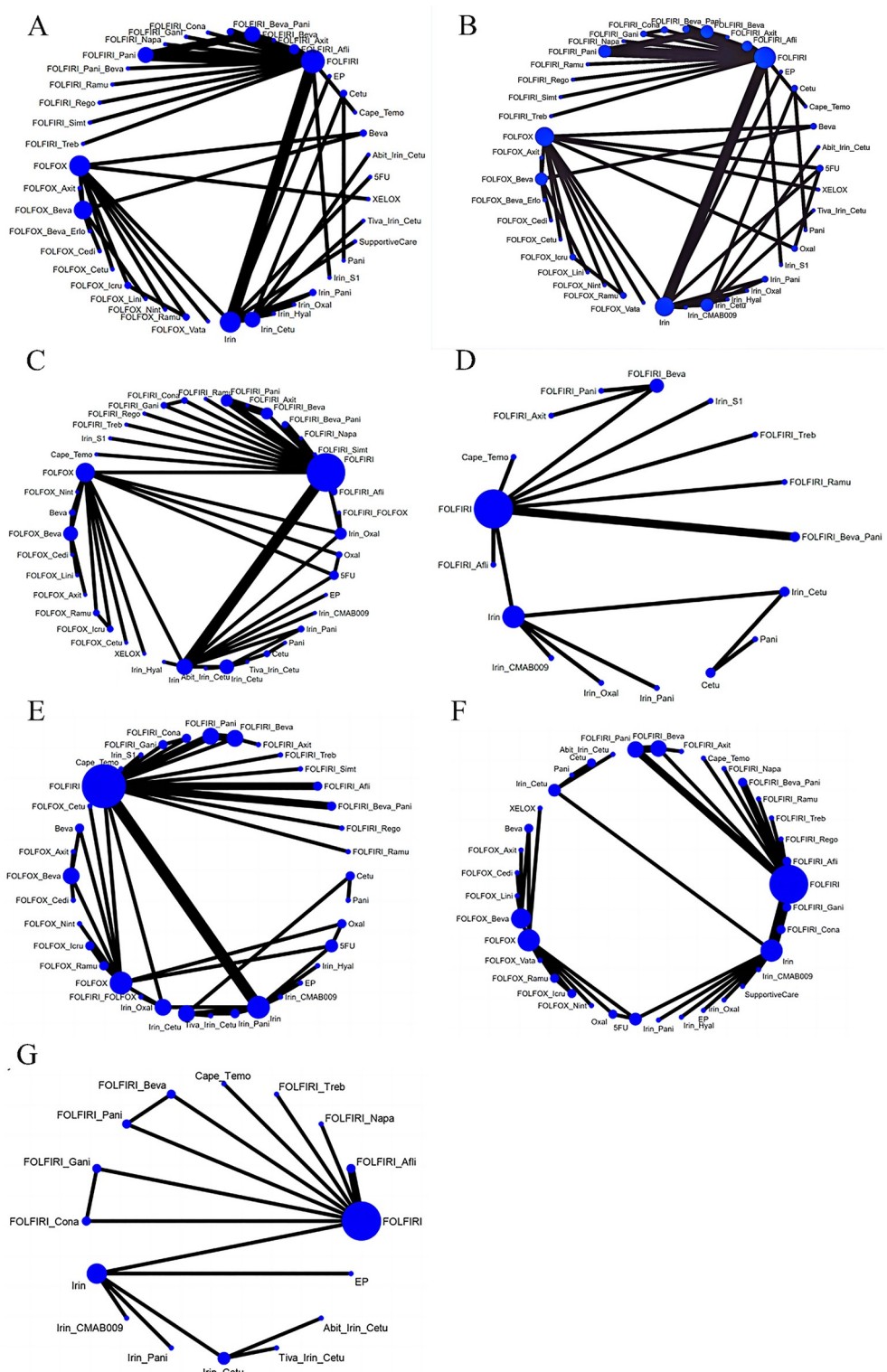

**Fig 2. Network diagram of available evidence.** (A) OS: Overall Survival; (B) PFS: Progression-Free-Survival; (C) ORR: Overall Response Rate; (D) CR: Complete Response; (E) PR: Partial Response; (F) Grade ≥ 3AE: Grade ≥ 3 Adverse Events; (G) Any AE: Any Adverse Events. Abit_Irin_Cetu: Abituzumab + Irinotecan + Cetuximab; Beva: Bevacizumab; Cape_Temo: Capecitabine + Temozolomide; Cetu: Cetuximab; EP: Etirinotecan Pegol; FOLFIRI_Afli: FOLFIRI + Aflibercept; FOLFIRI_Axit: FOLFIRI + Axitinib; FOLFIRI_Beva: FOLFIRI + Bevacizumab; FOLFIRI_Beva_Pani: FOLFIRI + Bevacizumab + Panitumumab; FOLFIRI_Cona: FOLFIRI + Conatumumab;

FOLFIRI_FOLFOX: FOLFIRI + FOLFOX; FOLFIRI_Gani: FOLFIRI + Ganitumab; FOLFIRI_Napa: FOLFIRI + Napabucasin; FOLFIRI_Pani: FOLFIRI + Panitumumab; FOLFIRI_Ramu: FOLFIRI + Ramucirumab; FOLFIRI_Rego: FOLFIRI + Regorafenib; FOLFIRI_Simt: FOLFIRI + Simtuzumab; FOLFIRI_Treb: FOLFIRI + Trebananib; FOLFOX_Axit: FOLFOX+Axitinib; FOLFOX_Beva: FOLFOX + Bevacizumab; FOLFOX_Beva_Erlo: FOLFOX + Bevacizumab + Erlotinib; FOLFOX_Cedi: FOLFOX + Cediranib; FOLFOX_Cetu: FOLFOX + Cetuximab; FOLFOX_Icru: FOLFOX + Icrucumab; FOLFOX_Lini: FOLFOX + Linifanib; FOLFOX_Nint: FOLFOX + Nintedanib; FOLFOX_Ramu: FOLFOX + Ramucirumab; FOLFOX_Vata: FOLFOX + Vatalanib; Irin: Irinotecan; Irin_Cetu: Irinotecan + Cetuximab; Irin_CMAB009: Irinotecan + CMAB009; Irin_Hyal: Irinotecan + Hyaluronan; Irin_Oxal: Irinotecan + Oxaliplatin; Irin_Pani: Irinotecan + Panitumumab; Irin_S1: Irinotecan + S-1; Oxal: Oxaliplatin; Pani: Panitumumab; SupportiveCare: Supportive Care; Tiva_Irin_Cetu: Tivantinib + Irinotecan + Cetuximab.

**3.6.3 Subgroup analysis.** Subgroup analyses of PFS and OS in mCRC patients were performed according to RAS status (RAS mutant or RAS wild-type).

Ten studies (with 1,957 patients) were included in the survival analysis of the RAS mutant population. According to the SUCRA probability rankings, in the OS, FOLFIRI + Bevacizumab + Panitumumab (SUCRA: 87.9%), FOLFIRI + Ramucirumab (SUCRA:57.1%), and FOLFIRI + Conatumumab (SUCRA: 56.7%) demonstrated best performance; while Etirinotecan pegol (SUCRA: 74.8%), Irinotecan + Panitumumab (SUCRA: 73.6%), and FOLFIRI + Bevacizumab + Panitumumab (SUCRA: 70.2%) demonstrated best performance regarding PFS. Fifteen studies containing 3954 cases were included in the survival analysis of the RAS wild-type population, and the results showed that FOLFIRI, VEGFR antagonists, and EGFR antagonists were more advantageous in prolonging OS. SUCRA values for FOLFIRI + bevacizumab, FOLFIRI + bevacizumab + panitumumab, and FOLFIRI + panitumumab were 73.2%, 65.8%, and 64.5%, respectively. Irinotecan + CMAB009 (SUCRA: 84.8%), FOLFIRI + Bevacizumab + Panitumumab (SUCRA: 83.0%), and Irinotecan + Panitumumab (SUCRA: 70.8%) demonstrated best performance regarding PFS (Table 4; Figs 7 and 8 in S1 File).

**3.6.4 Response.** According to the results of comprehensive comparison, 5-FU (RR = 0.02 [95%CI:0.00–0.30]), Bevacizumab (RR = 0.06[95%CI:0.00–0.90]), FOLFIRI (RR = 0.09[95% CI:0.00–0.75]), FOLFIRI + FOLFOX (RR = 0.04[95%CI:0.00–0.87]), FOLFIRI + Napabucasin (RR = 0.09[95%CI:0.00–0.92]), FOLFOX + Icrucumab (RR = 0.04[95%CI:0–0.80]), FOLFOX + Ramucirumab (RR = 0.04[95%CI:0–0.78]), Irinotecan (RR = 0.09[95%CI:0–0.84]) and Oxaliplatin (RR = 0.03[95%CI:0–0.50]) showed lower response rate compared with FOLFIRI + Trebananib, with statistically significant difference as shown in Fig 4C in S1 File. SUCRA ranking for probability indicated that the top three were FOLFIRI + Trebananib (SUCRA: 88.1%), FOLFIRI + Conatumumab (SUCRA: 87.2%) and FOLFOX + Bevacizumab (SUCRA: 81.1%) (Fig 3C; Table 3 in S1 File). In addition, Panitumumab (SUCRA: 79.6%) was the most competitive intervention in improving CR, with no significant difference compared with other regimens. The 5FU (RR = 0.03[95%CI:0–0.30]), Bevacizumab (RR = 0.08[95%CI:0–0.97]), Capecitabine + Temozolomide (RR = 0.06[95%CI:0–0.73]), FOLFIRI (RR = 0.10[95%CI:0–0.75]), FOLFIRI + FOLFOX (RR = 0.04[95%CI:0–0.73]), FOLFIRI + Ramucirumab (RR = 0.11 [95%CI:0–0.93]), FOLFIRI + Simtuzumab (RR = 0.09[95%CI:0–0.91]), FOLFOX + Icrucumab (RR = 0.05[95%CI:0–0.94]), FOLFOX + Ramucirumab (RR = 0.05[95%CI: 0–0.87]), Irinotecan (RR = 0.09[95%CI:0–0.74]), Irinotecan + S-1 (RR = 0.09[95%CI:0–0.77]), and Oxaliplatin (RR = 0.04[95%CI: 0–0.50]) showed lower PR compared with FOLFIRI + Conatumumab (SUCRA: 88.7%), with statistically significant difference (Fig 3D; 3E; Table 3; Fig 4D and 4E in S1 File).

**3.6.5 Adverse events.** The main reported indicator was Grade ≥ 3AE. Capecitabine + Temozolomide was correlated with lower Grade ≥ 3AE, but with no significant difference compared other interventions (Fig 4F in S1 File). It could be seen from the SUCRA ranking for probability, Capecitabine + Temozolomide was the least drug-related intervention

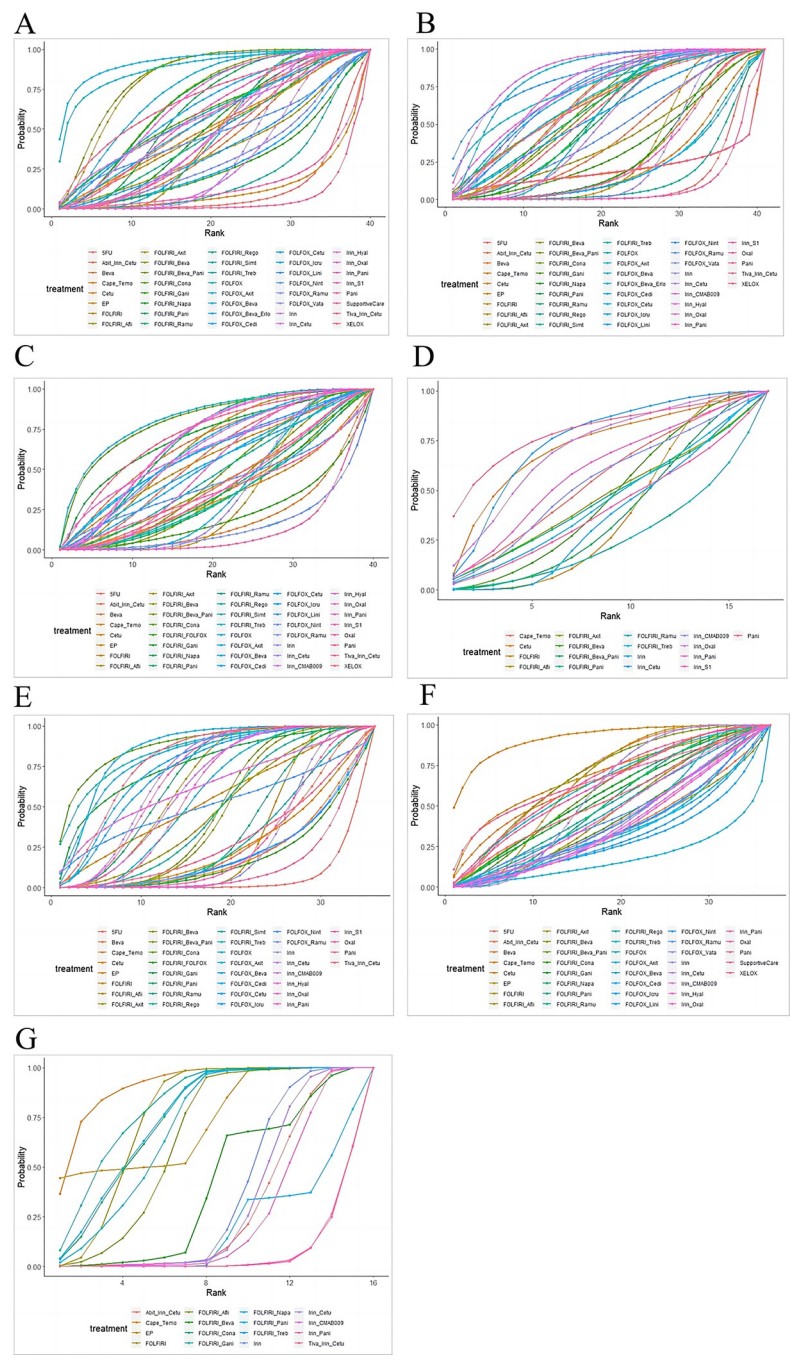

**Fig 3. The SUCRA diagrams of seven outcomes.** (A): Overall Survival; (B): Progression-Free-Survival; (C): Overall Response Rate; (D): Complete Response; (E): Partial Response; (F): Grade ≥3 Adverse Events; (G): Any Adverse Events.

(SUCRA: 91.9%), and Grade ≥ 3AE occurred the most times when FOLFOX + Axitinib was used (SUCRA: 17.9%) (Fig 3F; Table 3 in S1 File). According to the analysis of the Any AE occurrence, Capecitabine + Temozolomide (SUCRA: 91.4%) had the lowest occurrence rate of Any AE compared with Abituzumab + Irinotecan + Cetuximab (RR = 0.18[95%CI:0–0.83]), FOLFIRI + Bevacizumab (RR = 0.36[95%CI:0.03–0.90]), FOLFIRI + Panitumumab (RR = 0.08

[95%CI:0–0.54]), Irinotecan (RR = 0.19[95%CI:0–0.84]), Irinotecan + Cetuximab (RR = 0.18 [95%CI:0–0.83]), Irinotecan + CMAB009 (RR = 0.17[95%CI:0–0.78]), Irinotecan + Panitumumab (RR = 0.03[95%CI:0–0.47]), Tivantinib + Irinotecan + Cetuximab (RR = 0.03[95%CI:0– 0.46]), with statistically significant difference (Fig 3G; Table 3 in S1 File).

### 3.7 Sensitivity analysis and publication bias

Sensitivity analysis was based on the results of forest plots (Fig 5 in S1 File). The publication bias in the seven outcomes was assessed by developing funnel plots, and a rough distribution on both sides of the red line indicated small publication bias. The various interventions were represented by graphs with different colors, and most of the graphs in PFS, OS, CR, Grade $\geq$ 3AE, and Any AE were symmetrically distributed on both sides of the vertical line, indicating that there was no publication bias in these five indicators. The graphs of ORR and CR showed lower symmetry of distribution, indicating large publication bias (Fig 6 in S1 File).

## 4. Discussion

The goal of second-line treatment for mCRC is to prolong the survival and improve the quality of life of affected patients. Generally, second-line chemotherapy regimens include FOLFIRI, FOLFOX, and Irinotecan/Oxaliplatin single-agent chemotherapy. Second-line targeted agents mainly encompass VEGFR inhibitors, EGFR inhibitors, and agonist monoclonal antibodies human death receptor 5 (DR5). The choice of second-line therapeutic regimens depends on the first-line therapeutic regimen. Second-line irinotecan-based therapy is considered for patients with disease progression after first-line treatment containing olanzapine. FOLFOX/ XELOX alone or in combination is considered for patients with disease progression after first-line treatment containing irinotecan. First- and second-line VEGFR antagonists should be combined with EGFR antagonists [8]. In this study, 44 different second-line interventions for mCRC were included, and a comprehensive comparison and analysis of the regimens was performed in an attempt to provide evidence for clinical regimen selection.

Our study is the first to use Bayesian NMA to comprehensively explore the effectiveness (PFS, OS, ORR, CR, PR) and safety ($\geq$3 AE grade, any AE) of second-line treatment for mCRC based on all relevant published articles. The included articles were randomized controlled trials that enrolled sufficient patients in most treatment measure groups and grouped participants in a balanced manner. In this study, consistency testing and local inconsistency testing were also performed to make the analysis results more reliable. In addition, we performed subgroup analyses of survival data for the available RAS subtypes (RAS mutant and RAS wild-type) of the included patients.

According to the results of this study, treatments like Irinotecan + CMAB009, FOLFOX + Bevacizumab + Erlotinib, FOLFIRI + Trebananib, Panitumumab, FOLFIRI + Conatumumab, Capecitabine + Temozolomide, Capecitabine + Temozolomide ranked top regarding PFS, OS, ORR, CR, PR, Grade $\geq$ 3AE, and Any AE included for analysis, respectively, and the classic chemotherapy regimen FOLFOX had advantages over FOLFIRI in various outcome measures. For the RAS-mutant population, FOLFIRI + bevacizumab + panitumumab and Etirinotecan Pegol ranked highest in OS and PFS, respectively.

The survival outcome is a basic outcome that reflects the effectiveness of the drug. Compared with FOLFIRI/FOLFOX chemotherapy regimen alone, the combination strategy of FOLFOX with Bevacizumab or Axitinib significantly improved the efficacy and tumor response rate, and the combination strategy of FOLFIRI with Trebananib or Conatumumab was superior in improving the tumor response rate. Additionally, compared with Irinotecan as a single drug regimen, the addition of Panitumumab, Cetuximab and new EGFR antagonists

like CMAB009 also improved survival outcomes. Therefore, the combination of VEGF inhibitors or EGFR inhibitors with chemotherapy drugs can usually improve the prognosis of such patients. This result is consistent with the previous findings of a meta-analysis of first-line treatment options, which recommends FOLFOXIRI combined with Bevacizumab as the best treatment option for patients with mCRC [70]. In addition, molecular biomarkers are critical for the selection of second-line treatment options for mCRC. In mCRC, KRAS mutations occur in approximately 35–50% of patients and NRAS mutations in 3–5% of patients [71]. We performed a survival analysis of the RAS mutant and RAS wild-type populations. The results revealed that FOLFIRI + Bevacizumab + Panitumumab was effective in improving OS and PFS in the RAS-mutant population. Consistently, previous studies have shown that KRAS mutations predict ineffective treatment with EGFR inhibitors (panitumumab) [72], while combination with VEGFR antagonists proves to be beneficial [73]. In addition, the PRO-DIGE18 trial compared the efficacy of bevacizumab and cetuximab after disease progression in KRAS wild-type patients treated with bevacizumab-related first-line therapy, and it showed a survival advantage for bevacizumab, although the difference was not significant [47]. Notably, Etirinotecan Pegol, an irinotecan-related prodrug, showed encouraging results in enhancing PFS and could be applied in the future for the treatment of larger populations with RAS mutant phenotypes. In the RAS wild-type population, the combination regimen of FOLFIRI with bevacizumab (with or without an EGFR antagonist) ranked among the top in terms of enhancing both PFS and OS. Similarly, another NMA [74] included 15 original studies and concluded that the combination of chemotherapy and bevacizumab was the recommended second-line treatment for RAS wild-type mCRC patients according to the SUCRA ranking. In addition, 10 of the articles included in this study explicitly stated RAS mutants as the study population, and their first-line treatment regimens were not identical, so the data analysis could not be analyzed to derive the optimal treatment order for systemic therapy of mCRC in this population. However, based on the SUCRA ranking of each result, in both the general population and the RAS mutant population, the combination of chemotherapy and VEGFR antagonist can be recommended as an excellent regimen for patients with disease progression after first-line VEGFR/EGFR antagonist therapy.

In mCRC, the BRAF mutation rate is about 5–10% and about 90% are BRAFV600E mutations [75]. Mutations at this locus usually imply poor survival outcomes [76]. The PICCOLO study randomly assigned 460 KRAS wild-type patients who developed disease progression after first-line treatment to either the Irinotecan + panitumumab treatment group or the Irinotecan monotherapy group. Stratification according to BRAF revealed that the addition of panitumumab produced an unfavorable prognosis in patients with BRAFV600E mutations (HR 1.84, 95% CI 1.10–3.08) [56]. Zhang et al. collected data from five hospitals in China and randomly assigned patients to receive either FOLFIRI or Irinotecan, of which there were only four and three patients with the BRAFV600E mutation in the two groups, respectively. The result demonstrated that Irinotecan was superior to FOLFIRI in terms of both PFS and OS, but the results need to be interpreted with caution given the limited populations [67]. Both studies were included in this meta-analysis. Studies have shown that BRAF monotherapy is of low benefit as a second-line treatment for BRAF-mutated mCRC [77]. The BEACON study randomly assigned patients who received first-line treatments without BRAF, MEK, and EGFR to encorafenib, binimetinib plus cetuximab, binimetinib plus cetuximab or control treatment. The results showed that both triple therapy and double therapy significantly improved PFS, OS, and ORR. Encorafenib plus cetuximab with or without binimetinib was identified as a standard regimen for second- or third-line treatment in the mCRC population with BRAFV600E mutations [78]. The prognosis for mCRC patients with microsatellite highly unstable (MSI-H)/mismatch repair-deficient (dMMR) type is very poor, and they respond

poorly to common chemotherapy regimens [79]. The results of the Keynote 164 trial, a large multi-cohort non-randomized study, showed that as second-line and above treatment, Nivolumab in combination with low-dose Ipilimumab had an ORR of 65%, a PFS of 53%, 2-year OS of 71%, and manageable safety (serious AEs leading to discontinuation were seen in only 13% of patients) [78]. The latest findings showed that Nivolumab in combination with relatlimab had an ORR of 50%, 3-year PFS and OS of 38% and 56%, with only 8% of patients experiencing serious AEs leading to discontinuation [80]. Based on this study, Nivolumab has been recognized by the FDA as a recommended treatment for MSI-H patients.

Safety is also an important aspect in the assessment of pros and cons of second-line treatments for mCRC. According to the SUCRA rankings for probability in Grade $\geq$ 3AE or Any AE, Capecitabine + Temozolomide had a significant advantage, which might be related to the specific population included in this regimen (patients with RAS -mutated and MGMT-methylated mCRC). However, subgroup analysis of this population could not be performed with the small number of included patients in this regimen. Therefore, this result needed to be interpreted with caution [50]. It was worth mentioning that no significant difference was observed in the occurrence of adverse reactions between the various regimens according to the indirect comparison results, and therefore, the combination of drugs would improve the potential efficacy, but with possibly similar safety.

Overall, FOLFOX + Bevacizumab ranked top in the efficacy analysis with acceptable clinical tolerance. Based on the results of the subgroup analysis, chemotherapy + Bevacizumab (with or without Panitumumab) ranked higher in both OS and PFS. Therefore, the combination regimen of chemotherapy and bevacizumab was the treatment we preferred to recommend. This treatment was approved by the U.S. Food and Drug Administration (FDA) in 2007 as a second-line treatment for mCRC [81]. Bevacizumab is a humanized recombinant monoclonal antibody with high affinity for VEGF and can inhibit the binding of VEGF-A to the receptors VEGFR-1 and VEGFR-2 on vascular endothelial cells, thereby eliminating VEGF intracellular signal transduction and the biological outcome. In addition, Bevacizumab may improve the chemotherapy effect by changing tumor vasculature and reducing tumor interstitial pressure [82] and can significantly improve the prognosis of mCRC patients in the practice of second-line treatment of mCRC when combined with traditional chemotherapy FOLFOX. Common complications include nausea, vomiting, peripheral neuropathy, diarrhea, and neutropenia [25, 31, 35, 47, 63]. The research results showed that chemotherapy combined with Bevacizumab was superior to chemotherapy combined with Cetuximab [80], which was consistent with our results. In addition, Shi et al. conducted a RCT comparing FOLFOX + Bevacizumab with or without EGFR monoclonal antibody Erlotinib, and the results showed that the regimen combined with Erlotinib significantly prolonged PFS (9.6 months vs 6.9 months) with ORR of 48.5% vs 32.2%. However, the occurrence rate of adverse events was higher, and the number of patients included was limited, and this result needed to be verified by further research [62]. A previous large-scale systematic review compared chemotherapy combined with or without bevacizumab and concluded that the addition of bevacizumab significantly improved survival outcomes and tumor response rate without significantly increased incidence of serious complications. A traditional meta-analysis included the comparison of the combination treatments of molecule-targeted drugs and chemotherapy drugs in the second-line treatment of mCRC, and the results showed that anti-VEGF drugs had obvious advantages in improving survival [19]. Another traditional meta-analysis only included ten anti-VEGF drugs and revealed that chemotherapy combined with bevacizumab significantly improved PFS, OS, and ORR [16], which was similar to our results.

However, the NMA has some limitations. First, the open-label design of some studies led to a decrease in the quality of the included studies. Second, due to the lack of direct comparison,

some treatment strategies were excluded from the network analysis, especially in the subgroup analyses, which might have a certain impact on the results. Third, due to the inclusion criteria, this study failed to include patients with MSI mutants, and the included patients with BRAF mutants were too few to draw valid conclusions for this group of patients. Finally, although there were many included studies, the intervention measures were complicated, and each measure involved only a few studies, limiting the interpretation of our results.

## 5. Conclusions

In short, for the general population without RAS testing, FOLFOX + Bevacizumab was the optimum second-line treatment strategy for mCRC supported by our research results. For patients with RAS mutants, FOLFIRI + Bevacizumab + Panitumumab seemed to be an excellent option. However, the patient's physiological state was an important factor for the application of recommended treatment given that this treatment might have a certain number of serious complications. In this study, we analyzed the optimal second-line treatment regimen for mCRC patients without RAS stratification and for RAS-mutant mCRC patients. This NMA is expected to provide some evidence for clinical treatment and selection and may provide some ideas for future research. We expect more high-quality, multi-center RCT studies to supplement and validate this result.

## Supporting information

**S1 File. Basic support information.**
(DOCX)

**S2 File. Basic information for inclusion and exclusion of studies.**
(XLSX)

**S3 File. Data extraction table.**
(XLSX)

**S4 File. Quality assessment of the included studies.**
(XLSX)

**S5 File. PRISMA NMA checklist.**
(DOCX)

## Author Contributions

**Conceptualization:** Chengyu Sun.

**Formal analysis:** Chengyu Sun, Enguo Fan, Luqiao Huang, Zhengguo Zhang.

**Investigation:** Chengyu Sun, Enguo Fan, Luqiao Huang, Zhengguo Zhang.

**Methodology:** Chengyu Sun, Luqiao Huang.

**Resources:** Zhengguo Zhang.

**Supervision:** Zhengguo Zhang.

**Writing – original draft:** Chengyu Sun.

**Writing – review & editing:** Chengyu Sun, Enguo Fan.

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
