## [Decision Letter · Decision Letter 0]

30 Jul 2024

PONE-D-24-22429Second-line systemic treatment for metastatic colorectal cancer: A systematic review and Bayesian network meta-analysis based on RCTPLOS ONE

Dear Dr. Zhang,

Thank you for submitting your manuscript to PLOS ONE. After careful consideration, we feel that it has merit but does not fully meet PLOS ONE’s publication criteria as it currently stands. Therefore, we invite you to submit a revised version of the manuscript that addresses the points raised during the review process.

We look forward to receiving your revised manuscript.

Kind regards,

Md. Shahjalal

Academic Editor

PLOS ONE

 [This study has received funding by Technology Program of Xuzhou Municipal Health and Wellness Commission (KC21245)].  

4. PLOS requires an ORCID iD for the corresponding author in Editorial Manager on papers submitted after December 6th, 2016. Please ensure that you have an ORCID iD and that it is validated in Editorial Manager. To do this, go to ‘Update my Information’ (in the upper left-hand corner of the main menu), and click on the Fetch/Validate link next to the ORCID field. This will take you to the ORCID site and allow you to create a new iD or authenticate a pre-existing iD in Editorial Manager. Please see the following video for instructions on linking an ORCID iD to your Editorial Manager account: https://www.youtube.com/watch?v=_xcclfuvtxQ.

Additional Editor Comments (if provided):

Reviewers' comments:

Reviewer's Responses to Questions

**Comments to the Author**

1. Is the manuscript technically sound, and do the data support the conclusions?

Reviewer #1: Yes

Reviewer #2: Yes

2. Has the statistical analysis been performed appropriately and rigorously? 

Reviewer #1: Yes

Reviewer #2: Yes

3. Have the authors made all data underlying the findings in their manuscript fully available?

Reviewer #1: Yes

Reviewer #2: Yes

4. Is the manuscript presented in an intelligible fashion and written in standard English?

Reviewer #1: Yes

Reviewer #2: Yes

5. Review Comments to the Author

Reviewer #1: This is a meta-analysis examining second-line treatments for metastatic colorectal cancer (mCRC) using well-established protocols. Specifically, the authors used Bayesian network meta-analysis to systematically examine 47 randomized controlled trials (RCTs) that met their inclusion criteria, collected from multiple online research paper databases. These studies covered over 16,000 patients and 44 second-line treatments. The authors reported differences among different interventions and showed that FOLFOX + Bevacizumab was the overall best second line treatment. The study is well designed and clearly presented. The study provides important results that may guide clinical applications for mCRC.

Major comments:

This study is well designed and I have no major issues with the study. The only thing that I would recommend is to make clear the statistics whenever applicable, such as “P < 0.05, Wilcoxon rank-sum test”.

Minor comments:

1. Lns 54-56, please specify the estimation is for “global/world” or certain countries.

2. Lns 72-77, the introduction of second-line treatment should be consolidated into one sentence/clause.

3. Lns 84-85, for this concluding sentence, please specify that poor knowledge is for the treatments of mCRC.

4. Lns 187, replace “Due to” with “Because”.

5. Lns 196-198, this section is unacceptable. The authors should summarize the findings.

6. Ln 203, replace “and” with “or”.

7. Ln 306, replace “results” with “previous findings”.

8. Ln 327, replace “effect” with “outcome” or “efficacy”.

9. Lns 330-332, the two sentences should be consolidated into one.

10. Ln 371, replace “consideration” with “an important factor”.

Reviewer #2: The authors conducted a Bayesian network meta-analysis to establish a ranking of available treatments for progressing metastatic colon cancer. The systematic review and statistical analysis are well performed. However, several limitations constrain the generalizability of the findings. Following my major concernments:

• A primary concern is the absence of stratification factors for included trials. While the authors acknowledge this issue in lines 353-358, its implications are not adequately addressed. For second-line treatment of colorectal cancer, MSI and BRAF status are crucial determinants of optimal therapy. The methods section lacks explicit exclusion criteria for trials including MSI or BRAF-positive patients. Given the omission of nivolumab plus ipilimumab and encorafenib plus cetuximab from the network, clarification is needed on whether MSI and BRAF-positive patients were excluded. If so, this should be explicitly stated in the inclusion/exclusion criteria.

• Furthermore, the analysis overlooks the importance of treatment sequencing. Patients with RAS mutations may receive FOLFOX+Bevacizumab in the first-line setting. The study does not clarify whether FOLFOX+Bevacizumab (rechallenge?) remains the optimal second-line treatment for patients previously exposed to this regimen. This aspect warrants discussion and clarification.

• Considering the lack of stratification according to molecular information and previous treatments received, authors must discuss in the text how this work can help physicians to chose the best second line strategy for their patients.

Following some minor concernments:

• I suggest to better discuss the current scenario for II line treatment of metastatic colorectal cancer.

• Line 76: I suggest to change “sustaining” with “prolonging overall survival or progression free survival”.

• I suggest to summarize inclusion and exclusion criteria using PICOS model.

6. PLOS authors have the option to publish the peer review history of their article (what does this mean?). If published, this will include your full peer review and any attached files.

Reviewer #1: No

Reviewer #2: No

---

## [Author Response · Author response to Decision Letter 0]

4 Sep 2024

Thank you for your detailed review and comments on our manuscript entitled “Second-line systemic treatment for metastatic colorectal cancer: a systematic review and Bayesian network meta-analysis based on RCT” (ID: PONE-D-24-22429). These comments are of high value in revising and refining our paper and are important guidelines for our research. We have carefully considered your questions and have made corrections. Corrected sections are marked in red on the manuscript, and the main corrections and responses to you in the manuscript can be found below.

Reviewer #1:

This is a meta-analysis examining second-line treatments for metastatic colorectal cancer (mCRC) using well-established protocols. Specifically, the authors used Bayesian network meta-analysis to systematically examine 47 randomized controlled trials (RCTs) that met their inclusion criteria, collected from multiple online research paper databases. These studies covered over 16,000 patients and 44 second-line treatments. The authors reported differences among different interventions and showed that FOLFOX + Bevacizumab was the overall best second line treatment. The study is well designed and clearly presented. The study provides important results that may guide clinical applications for mCRC.

Major comments:

This study is well designed and I have no major issues with the study. The only thing that I would recommend is to make clear the statistics whenever applicable, such as “P < 0.05, Wilcoxon rank-sum test”.

Reply: Thank you for recognizing our manuscript and for your constructive comments. In this study, we did not use a p-value to determine whether there was a statistical difference, but rather whether the confidence interval contained 1. As per your suggestion, we have explained it in 2.6 Data analysis.

Minor comments:

1. Lns 54-56, please specify the estimation is for “global/world” or certain countries.

Reply: Thank you for your suggestions regarding the rigor of our data. We have made the following change “It is estimated that by 2030, the global burden of the disease will increase by 60%, with more than 2.2 million new cases and over 1.1 million deaths.”

2. Lns 72-77, the introduction of second-line treatment should be consolidated into one sentence/clause.

Reply: Thank you for your suggestion regarding our language structure issue. We have changed the sentence to “If a patient experiences disease progression or relapse during the treatment cycle but has good clinical performance and adequate organ function, he or she will require second-line therapy, which is used after first-line therapy due to progression or unacceptable toxicity, with the goal of prolonging overall survival or progression-free survival.”

3. Lns 84-85, for this concluding sentence, please specify that poor knowledge is for the treatments of mCRC.

Reply: Thank you for your careful review of our manuscript. We have changed it to “However, no up-to-date guidelines compare all available second-line treatments, and the optimal clinical second-line regimen is unclear.”

4. Lns 187, replace “Due to” with “Because”.

Reply: We appreciate your suggestion and have replaced “Due to” with “Because”.

5. Lns 196-198, this section is unacceptable. The authors should summarize the findings.

Reply: Thank you for your expert advice on our network diagrams. After careful consideration and interpretation, here we have clarified the number of mCRC second-line treatment options included in each outcome and have explained the rationale and results of network diagrams.

6. Ln 203, replace “and” with “or”.

Reply: We have replaced “and” with “or”. Thanks for your advice.

7. Ln 306, replace “results” with “previous findings”.

Reply: We have replaced “results” with “previous findings”. Thanks for your advice.

8. Ln 327, replace “effect” with “outcome” or “efficacy”.

Reply: We have replaced “effect” with “outcome”. Thanks for your advice.

9. Lns 330-332, the two sentences should be consolidated into one.

Reply: Thank you for your suggestion. There is a semantic duplication here. We have changed it to “Common complications include nausea, vomiting, peripheral neuropathy, diarrhea, and neutropenia.”

10. Ln 371, replace “consideration” with “an important factor”.

Reply: We have replaced “consideration” with “an important factor”. Thanks for your advice.

Reviewer #2:

The authors conducted a Bayesian network meta-analysis to establish a ranking of available treatments for progressing metastatic colon cancer. The systematic review and statistical analysis are well performed. However, several limitations constrain the generalizability of the findings. Following my major concernments:

• A primary concern is the absence of stratification factors for included trials. While the authors acknowledge this issue in lines 353-358, its implications are not adequately addressed. For second-line treatment of colorectal cancer, MSI and BRAF status are crucial determinants of optimal therapy. The methods section lacks explicit exclusion criteria for trials including MSI or BRAF-positive patients. Given the omission of nivolumab plus ipilimumab and encorafenib plus cetuximab from the network, clarification is needed on whether MSI and BRAF-positive patients were excluded. If so, this should be explicitly stated in the inclusion/exclusion criteria. 

Reply: Thank you for your questions and constructive comments. Regarding the stratification factors included in the study, what we did previously was indeed not comprehensive enough, so we have made the following corrections:

(1) We again carefully reviewed the literature screening process and confirmed that no trials with MSI- and BRAF-positive patients were excluded based on the inclusion criteria.

(2) A careful reading and data extraction of all included studies revealed no MSI gene stratification data available for analysis, and only 3 studies performed stratification according to BRAF gene status. Therefore, it was not possible to analyze patient data for MSI and BRAF status. However, based on your valuable comments, we carefully discussed the recommended treatments for MSI and BRAF-positive patients (lines 385 through 412). In addition, we carefully reviewed RAS gene status, another stratification factor clinically closely related to the choice of mCRC treatment regimen. We identified 22 studies that stratified for RAS gene status. A total of 1,957 patient data from 10 studies were available for the analysis of OS and PFS in RAS mutant patients; 3,954 patient data from 15 studies were available for the analysis of OS and PFS in RAS wild-type patients. We therefore performed subgroup analyses for this (lines 245 to 262; S7 Fig; S8 Fig; S4 Table). The results showed that the FOLFIRI + Bevacizumab + Panitumumab regimen was recommended in the RAS mutant population. It is also carefully described in the Discussion and Conclusions.

• Furthermore, the analysis overlooks the importance of treatment sequencing. Patients with RAS mutations may receive FOLFOX+Bevacizumab in the first-line setting. The study does not clarify whether FOLFOX+Bevacizumab (rechallenge?) remains the optimal second-line treatment for patients previously exposed to this regimen. This aspect warrants discussion and clarification.

Reply: Thank you for your constructive comments. Based on your valuable comments, we collected and analyzed data according to RAS gene status. A total of 11 studies with RAS genotyping data were collected (S2 Table), and data from 10 studies were available for analysis (the second-line regimen compared in the study by Moore et al. could not be linked to the second-line regimens of the other studies). Since their first-line treatment regimens were not identical, the optimal treatment order could not be derived from this analysis. However, according to the SUCRA ranking, chemotherapy combined with bevacizumab combined with an EGFR antagonist performed well in the RAS mutant population. Therefore, we recommend FOLFIRI + bevacizumab + panitumumab for those using FOLFOX + bevacizumab as the first-line treatment. This has been described in detail in the subgroup analysis of the results and in the discussion.

• Considering the lack of stratification according to molecular information and previous treatments received, authors must discuss in the text how this work can help physicians to chose the best second line strategy for their patients.

Reply: Thank you for your insights and valuable comments. We have supplemented the subgroup analysis of RAS genotyping and based on the results, we recommend the FOLFIRI + bevacizumab + panitumumab regimen as the second-line treatment in the RAS-mutant population, and it remains applicable in the population who have used FOLFOX + bevacizumab as the first-line treatment. In addition, although this study did not derive the optimal second-line treatment for the MSI and BRAF-positive populations, the optimal second-line regimens for these two populations have been discussed and integrated with the authoritative literature. Therefore, our study has recommended the optimal second-line treatment for mCRC patients based on different populations, which is described in detail in the conclusion and discussion. Thank you again for your valuable comments on treatment order and stratification factors, so that we can do our best to refine our results to make this study results better applied in clinical practice.

Following some minor concernments:

• I suggest to better discuss the current scenario for II line treatment of metastatic colorectal cancer.

Reply: Thank you for your valuable comments. We have discussed the current status of second-line treatment options in more detail in the first paragraph of the discussion.

• Line 76: I suggest to change “sustaining” with “prolonging overall survival or progression free survival”.

Reply: We have replaced “sustaining” with “prolonging overall survival or progression free survival”. Thanks for your advice.

• I suggest to summarize inclusion and exclusion criteria using PICOS model.

Reply: We appreciate your suggestion. We have summarized the inclusion and exclusion criteria using the PICOS model. The revised section is shown below:

Studies that met the following criteria were included:

(1) Population: the subjects of the study were mCRC patients; (2) Intervention and control: a second-line treatment was used in both the study and control groups, and the drug types, dosages, and order of medication must be specified for the second-line and first-line treatment regimens; (3) Outcome: the reported results included PFS, overall survival (OS), overall response rate (ORR), complete response (CR), partial response (PR), grade 3 and above adverse events (Grade ≥ 3AE), and some or all of any adverse events (Any AE); (4) Study design: the study type was a randomized controlled trial. If more than one article reported the same population, the one with more complete data was included.

---

## [Decision Letter · Decision Letter 1]

22 Oct 2024

Second-line systemic treatment for metastatic colorectal cancer: A systematic review and Bayesian network meta-analysis based on RCT

PONE-D-24-22429R1

Dear Dr. Zhang,

We’re pleased to inform you that your manuscript has been judged scientifically suitable for publication and will be formally accepted for publication once it meets all outstanding technical requirements.

Kind regards,

Md. Shahjalal

Academic Editor

PLOS ONE

Additional Editor Comments (optional):

Reviewers' comments:

Reviewer's Responses to Questions

**Comments to the Author**

1. If the authors have adequately addressed your comments raised in a previous round of review and you feel that this manuscript is now acceptable for publication, you may indicate that here to bypass the “Comments to the Author” section, enter your conflict of interest statement in the “Confidential to Editor” section, and submit your "Accept" recommendation.

Reviewer #1: All comments have been addressed

Reviewer #3: All comments have been addressed

2. Is the manuscript technically sound, and do the data support the conclusions?

Reviewer #1: Yes

Reviewer #3: Yes

3. Has the statistical analysis been performed appropriately and rigorously? 

Reviewer #1: Yes

Reviewer #3: Yes

4. Have the authors made all data underlying the findings in their manuscript fully available?

Reviewer #1: Yes

Reviewer #3: No

5. Is the manuscript presented in an intelligible fashion and written in standard English?

Reviewer #1: Yes

Reviewer #3: Yes

6. Review Comments to the Author

Reviewer #1: Thanks for addressing my comments. The revision has substantial improvements. I recommend acceptance of this paper.

Reviewer #3: (No Response)

7. PLOS authors have the option to publish the peer review history of their article (what does this mean?). If published, this will include your full peer review and any attached files.

Reviewer #1: **Yes: **Qian Xu

Reviewer #3: No

---

## [Editor Report · Acceptance letter]

10 Dec 2024

PONE-D-24-22429R1 

PLOS ONE

Dear Dr. Zhang, 

I'm pleased to inform you that your manuscript has been deemed suitable for publication in PLOS ONE. Congratulations! Your manuscript is now being handed over to our production team.

Kind regards, 

on behalf of

Dr. Md. Shahjalal 

Academic Editor

PLOS ONE